# A RAS-Independent Biomarker Panel to Reliably Predict Response to MEK Inhibition in Colorectal Cancer

**DOI:** 10.3390/cancers14133252

**Published:** 2022-07-01

**Authors:** Ulrike Pfohl, Jürgen Loskutov, Sanum Bashir, Ralf Kühn, Patrick Herter, Markus Templin, Soulafa Mamlouk, Sergei Belanov, Michael Linnebacher, Florian Bürtin, Marcus Vetter, Christoph Reinhard, Lena Wedeken, Christian R. A. Regenbrecht

**Affiliations:** 1CELLphenomics GmbH, Robert-Rössle-Str. 10, 13125 Berlin, Germany; ulrike.pfohl@cellphenomics.com (U.P.); juergen.loskutov@cellphenomics.com (J.L.); christoph.reinhard@cellphenomics.com (C.R.); lena.wedeken@cellphenomics.com (L.W.); 2Institute for Molecular Biosciences, Goethe University Frankfurt am Main, Max-von-Laue-Str. 13, 60438 Frankfurt am Main, Germany; 3Genome Engineering & Disease Models, Max Delbrück Center for Molecular Medicine, Robert-Rössle-Str. 10, 13125 Berlin, Germany; sanum.bashir@biontech.de (S.B.); ralf.kuehn@mdc-berlin.de (R.K.); 4NMI Natural and Medical Sciences Institute, University of Tübingen, Markwiesenstraße 55, 72770 Reutlingen, Germany; patrick.herter@boehringer-ingelheim.com (P.H.); markus.templin@nmi.de (M.T.); 5ASC Oncology GmbH, Robert-Rössle-Str. 10, 13125 Berlin, Germany; 6Institute of Pathology, Charité Universitätsmedizin Berlin, Virchowweg 15, Charitépl. 1, 10117 Berlin, Germany; soulafa.mamlouk@charite.de; 7Institute of Biotechnology, University of Helsinki, Viikinkaari 5, Biocenter 2, 00790 Helsinki, Finland; sergei.belanov@helsinki.fi; 8Clinic of General Surgery, Molecular Oncology and Immunotherapy, Rostock University Medical Center, Schillingallee 35, 18057 Rostock, Germany; michael.linnebacher@med.uni-rostock.de; 9Clinic of General Surgery, Rostock University Medical Center, Schillingallee 35, 18057 Rostock, Germany; florian.buertin@med.uni-rostock.de; 10University Hospital Basel, Petersgraben 4, 4031 Basel, Switzerland; marcus.vetter@ksbl.ch; 11Kantonsspital Baselland, Rheinstr. 26, 4410 Liestal, Switzerland; 12Institute of Pathology, University Hospital Göttingen, Robert-Koch-Straße 40, 37075 Göttingen, Germany

**Keywords:** organoids, biomarker, targeted therapy, colorectal cancer, CRC, *SMAD4*, TGF-β/BMP-pathway, intra-tumor heterogeneity, MEK inhibition

## Abstract

**Simple Summary:**

Today, clinical management for the majority of cancer patients is still based on a “one-size-fits-all” approach. To improve the outcomes in the era of personalized medicine, it is essential to stratify patients based on established and novel biomarkers. In the present study, we investigated a *SMAD4* loss-of-function mutation, which is associated with chemoresistance and decreased overall survival in colorectal cancer (CRC). To investigate the molecular mechanism behind the impact on drug response, we used CRISPR technology on patient-derived organoid models (PDOs) of CRC. We showed that PDOs with loss-of-function *SMAD4* mutations are sensitive to MEK-inhibitors. Using a novel four-gene signature reliably predicts sensitivity towards MEK-inhibitors, regardless of the *RAS* and *BRAF* status. The present study is a significant step towards personalized cancer therapy by identifying a new biomarker.

**Abstract:**

**Background:** In colorectal cancer (CRC), mutations of genes associated with the TGF-β/BMP signaling pathway, particularly affecting *SMAD4*, are known to correlate with decreased overall survival and it is assumed that this signaling axis plays a key role in chemoresistance. **Methods:** Using CRISPR technology on syngeneic patient-derived organoids (PDOs), we investigated the role of a loss-of-function of *SMAD4* in sensitivity to MEK-inhibitors. CRISPR-engineered *SMAD4*^R361H^ PDOs were subjected to drug screening, RNA-Sequencing, and multiplex protein profiling (DigiWest^®^). Initial observations were validated on an additional set of 62 PDOs with known mutational status. **Results:** We show that loss-of-function of *SMAD4* renders PDOs sensitive to MEK-inhibitors. Multiomics analyses indicate that disruption of the BMP branch within the TGF-β/BMP pathway is the pivotal mechanism of increased drug sensitivity. Further investigation led to the identification of the SFAB-signature (*SMAD4*, *FBXW7*, *ARID1A*, or *BMPR2*), coherently predicting sensitivity towards MEK-inhibitors, independent of both *RAS* and *BRAF* status. **Conclusion:** We identified a novel mutational signature that reliably predicts sensitivity towards MEK-inhibitors, regardless of the *RAS* and *BRAF* status. This finding poses a significant step towards better-tailored cancer therapies guided by the use of molecular biomarkers.

## 1. Introduction

Colorectal cancer (CRC) is the third most diagnosed cancer in the world [1]. The main cause of death in CRC is metastases [2]. At first diagnosis, 20% of colorectal cancer patients already have developed metastases, and more than 30% of early-stage patients will eventually develop metastases [3]. Making a successful treatment even more difficult, the metastatic disease is often refractory to systemic agents such as 5-fluorouracil (5-FU)-based therapies [4,5], so progression-free survival after first-line treatment did not significantly improve in the last decade [6,7]. The main reason for this unsatisfactory situation is that large cohorts of patients may benefit from guideline treatments, yet the individual tumors are so heterogenous that for most metastatic CRC patients, this “one-size-fits-all” approach does not culminate in a successful treatment [8,9]. This fact is fully accepted by the approving bodies (EMA, FDA); therefore, the development of companion diagnostic biomarkers is highly encouraged for future targeted treatment regimens.

Loss-of-function mutations of the *SMAD4* gene are present in up to 15% of CRC cases [10]. They have been correlated with decreased overall survival [11,12] and a strong line of evidence suggests that functional loss of *SMAD4* promotes chemoresistance in multiple cancer types, including CRC [13].

In the present study, we link *SMAD4* loss-of-function mutations and the subsequent loss of the TGF-β/BMP signaling to increased sensitivity towards MEK-inhibitors of affected cells. More specifically, the BMP axis of this signaling pathway seems responsible for resistance towards MEK inhibition. This led to the identification of a mutational signature present in more than 30% of CRC cases as sufficient for predicting sensitivity to MEK-inhibitors independent of the *RAS* and *BRAF* mutational status of the tumor.

## 2. Materials and Methods

### 2.1. 3 D Cell Culture

PDO cultures were generated and propagated as previously described [14,15]. Models used are listed in Appendix A.

### 2.2. Semi-Automated High-Throughput Drug Response Assay

PDO cultures were digested with TrypLE Express (ThermoFisher Scientific, Waltham, MA, USA) to a single cell suspension. Single cells were recovered in 5 mL complete PDO medium with ROCK-inhibitor (Miltenyi Biotec, Bergisch Gladbach, Germany) for 15 min at 37 °C. Cells were embedded in Matrigel^®^ (Corning, Tewksbury, MA, USA) at a density of 1500 cells/well in 384-well plates (Greiner Bio-One, Frickenhausen, Germany) using a Biomek FX automated workstation (Beckman Coulter, Brea, CA, USA). Plates were incubated for 15 min at 37 °C to allow Matrigel^®^ solidification, and a 50 µL/well complete PDO medium with ROCK-inhibitor was added. PDOs were cultured until organoid sizes reached 50 µm in diameter prior to treatment. Subsequently, drug treatments were performed by removing the medium daily and adding 50 µL/well complete medium without ROCK-inhibitor, containing drugs in 4 replicates. Concentrations of the tested compounds were distributed around the c_max_ (maximal plasma concentration) values observed in clinical settings. Cells treated with DMSO (Sigma-Aldrich, Steinheim, Germany) (0.3%) were used as “untreated control,” while Matrigel^®^ only was used for background readout. Staurosporine was used as an internal positive control.

After 4-day treatment, the cell viability was evaluated by Cell Titer Glo^®^ (CTG) assay (Promega, Madison, WI, USA) following the manufacturer’s protocol. The intensity of luminescence was measured using the SpectraMax i3x plate reader (Molecular Devices, San Jose, CA, USA) 30 min after the addition of the reagent. Luminescence intensity data were used to evaluate the percentage of viability after background subtraction. To interpret the data, organoids are defined as sensitive if the absolute IC_50_ value is equal to or lower than the c_max_ value and resistant if the absolute IC_50_ value is higher than the c_max_ value.

### 2.3. Genomic DNA and RNA Isolation

For DNA preparation, PDOs from 4–8 wells of a 24-well plate were collected, centrifuged (5 min, 300× *g*), washed in 1× PBS (ThermoFisher Scientific), and stored at −80 °C for further use. Genomic DNA was isolated using the Wizard Kit (Promega) according to the manufacturer’s protocol. DNA was eluted in 30 µL RNAase-free water (Sigma-Aldrich) and stored at −20 °C for further use.

For total RNA isolation, PDOs from 12 wells of a 24-well plate were collected. PDOs were centrifuged (300× *g* for 5 min at 4 °C) and kept on ice for the following steps. Pellets were washed with 5 mL sterile ice-cold PBS and centrifuged. Cell pellets were resuspended in 600 µL ice-cold RLT buffer (Qiagen, Venlo, The Netherlands) supplemented with 1% β-mercaptoethanol. Samples were frozen at −80 °C for further use. Total RNA was isolated by ATLAS Biolabs GmbH (Berlin, Germany). Nucleic acids were quantified using the Nanodrop 2000 (ThermoFisher Scientific).

### 2.4. PCR

Primers were designed using the web-interface Primer3Plus. The PCR reaction mixture (25 μL) contained 100 ng isolated DNA, 1× Phusion GC buffer, 3% DMSO, 200 μM dNTPs, 0.5 μM of each primer (SMAD4_fw 5′-AAATTCTCAGTTGACCTGGTCC-3′ and SMAD4_rev 5′-ACCGACAATTAAGATGGAGTGC-3′), and 0.02 U/μL of Phusion DNA polymerase (all PCR reagents from ThermoFisher Scientific). Initial denaturation at 95 °C for 5 min was followed by 34 cycles of denaturation at 95 °C for 30 s, primer annealing at 58 °C for 30 s, and extension at 72 °C for 30 s in an automated DNA thermal cycler (Biometra TRIO Thermal Cycler Series, Analytik Jena AG, Jena, Germany). The final extension was performed at 72 °C for 5 min. The PCR products, expected to be 688 bp, were resolved on 2% agarose gel stained with ethidium bromide (Roth, Karlsruhe, Germany) and visualized under the UV trans illuminator (VWR^®^ International, Radnor, PA, USA). Molecular size markers (GeneRulerTM100bp DNA Ladder, ThermoFisher Scientific) were run concurrently. PCR products were purified using the QIAquick PCR purification Kit (Qiagen), according to the manufacturer’s protocol.

### 2.5. CRC Panel Sequencing (DNAseq)

Targeted high-depth sequencing was performed for genomic areas comprising 100 genes using a custom-designed CRC panel (Appendix A) [16]. From each sample, 10 ng of DNA were used for multiplexed PCR amplification with the Ion Ampliseq Library kit (ThermoFisher Scientific), using two amplicon pools per DNA sample. Samples were ligated to Ion Xpress Barcode Adapters (ThermoFisher Scientific) and purified using Agencourt AMPure beads (Beckman Coulter). All six samples were combined on a 316v2 chip and sequenced on an Ion Torrent PGM device (ThermoFisher) with an average read depth of 580–700.

### 2.6. Sanger Sequencing

Mutations found via panel sequencing were validated via direct Sanger sequencing of PCR amplicons using both forward and reverse *SMAD4* PCR primers mentioned before. Aliquots of 14 µL 20 ng/µL purified PCR product were sent to LGC Genomics GmbH (Berlin, Germany) for sequencing. Electropherograms and FASTA files from Sanger sequencing were analyzed internally using CLC Genome Viewer (Qiagen).

### 2.7. Whole Transcriptome Sequencing (RNAseq)

PDOs were treated with trametinib (0.03 µM) or vehicle control DMSO (0.3%) for 24 h. Drug concentrations were chosen based on clinically achievable plasma concentrations [17]. RNA isolation and whole-transcriptome sequencing were performed at ATLAS Biolabs GmbH. An average of 1 μg total RNA per sample was used to generate barcode-labeled libraries using the Illumina TruSeq Standard mRNA Library preparation kit. Paired-end libraries were sequenced on an Illumina NovaSeq sequencer with the 125 bp reads protocol. The quality of raw RNA-seq reads was estimated with the following trimming of adapters and low-quality reads with the fastP [18] and MultiQC tools [19]. To generate gene expression data, RNA-seq reads were mapped to the latest version of Homo sapiens GRCh37 reference genome from the Ensembl collection [20] using STAR aligner [21]. Rsubread package [22] and featureCounts tool [23] were used to evaluate read counts and generate the expression data matrix. Further differential gene expression analysis was performed using the NOISeq package [24] with the pipeline customized for the replicate simulation analysis. Read counts were normalized using RPKM (Reads per kilobase per million mapped reads) [25] and TMM (Trimmed Mean of M-values) methods [26], low count values were filtered away, and the ARSyNseq function of the NOISeq package was used to filter out noise-associated results and unidentified batch effects. To increase the significance of differential expression probability, we set parameter q = 0.95 for the NOISeq-sim function. Functional enrichment analysis was done using parameter q = 0.9 to have a more representative and broader picture of affected pathways. Bioconductor packages org.Hs.eg.db (https://doi.org/10.18129/B9.bioc.org.Hs.eg.db, accessed on 13 October 2021) and Genomic Features [27] were used to annotate the results. GO-annotation and functional enrichment analysis was implemented utilizing GOexpress [28] and EnrichR [29,30] packages.

### 2.8. Generating CRISPR-Engineered Organoids

#### 2.8.1. Cloning Procedure

*SMAD4* single-guided RNA (sgRNA) sgSMAD4-i-A (5′-CACCGATGGATACGT-GGACCCTTC-3′) was inserted into the BbsI sites of pU6chimRNA-CAG-Cas9-venus-bpA (kindly provided by Ralf Kühn) (Appendix A). The plasmid was digested with BbsI (NEB #R3539) for 1.5 h at 37 °C. The mixture of linearized plasmid and inserts were incubated with T4 DNA ligase (NEB #M0202L) overnight at 16 °C, followed by transformation into *E. coli*. DH5*α*. Plasmid minipreps were performed according to the standard protocol (GeneJET Plasmid Miniprep Kit, Thermo Fisher Scientific). The resulting plasmid (pU6-SMAD4-I-CAG-Cas9-venus-bpA) was sequenced using U6 sequencing primer 5′-GAGGGCCTA-TTTCCCATG-3′ and was further used for transfection. All primers and single-stranded oligonucleotides were synthesized by BioTEz (Berlin, Germany).

#### 2.8.2. Transfection for Targeted Gene Editing

*SMAD4*^wt^ PDOs were collected and digested as previously described. 2 × 10^5^ single cells were plated into 24-well ULA plates (Greiner Bio-One) in 500 µL of complete PDO medium 6 h before transfection. Transfection was performed with TransIT-2020 (Mirus Bio, Madison, WI, USA), according to the manufacturer’s protocol. Single-stranded oligos (ssODNs) were used for the introduction or correction of single nucleotide mutation via homology-directed repair at the target site. Cells were transfected with 700 ng DNA MasterMix, consisting of 250 ng pU6SMAD4-I-CAG-Cas9-venus-bpA (Appendix A), 200 ng pCAG-Cas9-PCK-venus, 166.6 ng ssODN (5′-ACCTTGCTCTCTCAATGGCTTCTGTCCT-GTGGACATTGGAGAGTTGACCCAAACAAAAGCGATCTCCTCCAGAAGGGTCCACGTATCCATCAACAGTAACAATAGGGAGCTTGAAGGA-3′) (Appendix A) as repair template bearing the R361H point mutation (Appendix A) and 83.3 ng pCAG-i53-EF1BFP (Appendix A) to increase homologous-directed repair [31].

#### 2.8.3. Cell Sorting

72 h after transfection, cells were collected, centrifuged at 300× *g* for 5 min, and digested in 500 µL TrypLE Express for 15 min. Single cells were washed in a complete PDO medium with ROCK-inhibitor and incubated for 15 min at 37 °C, washed in PBS, and filtered through a 35 µm cell strainer (Corning). Single cells were gated for BFP/Venus and sorted directly in 500 µL of complete PDO medium supplied with 10 µg/mL gentamycin (Lonza, Basel, Switzerland) using a BD Aria cell sorter (BD Biosciences, Franklin Lakes, NJ, USA). BFP/Venus-positive cells were embedded in Matrigel^®^. After 3 weeks in culture, one-half of the cells was used for isolation of genomic DNA and performing genotyping PCR, while the other was used for expansion and clonal selection.

#### 2.8.4. Limited Dilution

Cells were dissociated using TrypLE Express as previously described. A cell solution at a concentration of 5 cells/mL was prepared in a 10 mL complete PDO medium. 100 μL per well were transferred into a round 96-well plate (Greiner Bio-One) to achieve an average density of 0.5 cells/well. Individual colonies were allowed to grow until they could be genotyped individually to confirm gene editing using PCR and the pJet cloning method (Thermo Fisher Scientific #EF694056.1).

### 2.9. High-Content Western Blotting–DigiWest^®^

DigiWest^®^ on PDO cultures was performed essentially as previously described [32]. For organoid generation, 3.0 × 10^6^ cells per condition were plated. After 72 h, PDOs were treated with trametinib (0.03 μM) or vehicle control DMSO (0.3%) for 24 h. PDOs were isolated with cell recovery solution (Corning) according to the manufacturer’s protocol and stored at −80 °C. For analysis, organoid pellets were lysed by the addition of 2× LDS lysis buffer (Life Technologies) containing Protease-Inhibitor Mix M (Serva, Heidelberg, Germany), PhosSTOP (Roche Applied Science, Mannheim, Germany), and PMSF (ThermoFisher Scientific) and heating to 95 °C for 10 min; protein was quantified using a Pierce BCA Protein Assay Kit (ThermoFisher Scientific). The NuPAGE SDS-PAGE gel system (Thermo Fisher) was used for protein separation and blotting; 12 µg protein per sample was separated using 4–12% Bis-Tris gels and transferred onto PVDF membranes (Millipore, Billerica, MA, USA).

For the DigiWest^®^ procedure, relative protein expression was determined using the primary antibodies listed in Appendix A and signal intensity was determined from the integral of the analyte-specific peak. The values were normalized to beta-actin and data were presented relative to the DMSO controls.

### 2.10. Establishment of 5-FU Resistant PDO Models

PDO models were cultivated as described before. To establish 5-FU resistant PDO models, cells were grown in the presence of a sublethal concentration (9.2 µM) of 5-FU for 4 weeks. Surviving cells were allowed to recover and challenged further with 46.1 µM of 5-FU over 4 months. Subsequently, 5-FU concentration was increased to 230 µM (c_max_) [33] for another 3 months. Before drug screening, 5-FU resistant PDOs, as well as the non-resistant counterparts, were cultivated and passaged in a complete PDO medium. 5-FU resistant PDOs were treated with 5-FU, regorafenib, cobimetinib, and trametinib and compared with their non-resistant counterparts.

## 3. Results

### 3.1. SMAD4 Loss-of-Function Is Associated with Differential Drug Response

Intra-tumor heterogeneity poses a significant obstacle in cancer therapy. However, comparing different cell populations from the same tumor can be used to understand mechanistic complexity and subsequently identify prospective biomarkers. We have previously reported that PDO “sibling”-culture models established from separate regions of a primary chemo-naïve CRC tumor respond differently to drugs [15]. Here, we focused on two of these sibling PDO models (R1^R361H^ and R4^wt^), displaying multiple identical mutations (*KRAS*^G12D^, *PIK3CA*^H1047R^, and *TP53*^C242F^), but differ regarding a *SMAD4*^R361H^ mutation (Figure 1A). As reported by others [34], PDO model R1^R361H^ carries the well-known loss-of-function mutation *SMAD4*^R361H^ [35] and tends to proliferate faster in comparison to R4^wt^ (Figure 1B).

There were no significant differences between the two PDOs in response to 5-FU, a pivotal drug in first-line therapy for CRC (Figure 1C). Notably, the absolute IC_50_ values for both models were very close to the c_max_ values of 5-FU [33], reflecting the donor patient’s poor response to FOLFOX/FOLFIRINOX in the clinic. Morphological changes observed in R1^R361H^ upon 5-FU treatment (Appendix A) may signify epithelial to mesenchymal transition (EMT). In contrast, responses to targeted therapeutics, however, revealed drastic differences in sensitivity to MEK-inhibitors cobimetinib, trametinib, and selumetinib. The mutant PDO R1^R361H^ was 10- to 100-fold more sensitive towards the MEK-inhibitors trametinib and cobimetinib, respectively, compared to the *SMAD4*^wt^ counterpart R4^wt^ (Figure 1E and Appendix A). Whereas targeting the mitogen-activated protein kinase (MAPK) pathway upstream or downstream (Figure 1D) of MEK resulted in similar values in both models (Figure 1E and Appendix A). Thus, the *SMAD4* mutational status is very likely responsible for a differential sensitivity specifically to MEK inhibition.

### 3.2. Introduction of SMAD4 Point Mutation Reverses MEK-Inhibitor Resistance

To substantiate our hypothesis that loss-of-function *SMAD4* mutations create sensitivity to MEK-inhibitors, we generated syngeneic PDOs harboring *SMAD4*^R361H^. The R4^wt^ PDO model was CRISPR-engineered to introduce a single *SMAD4*^R361H^ mutation (Figure 2A and Appendix A). Three such clones (clones A, B, and C) were established and the successful introduction of the *SMAD4*^R361H^ mutation was confirmed (Figure 2B,C, Appendix A). *SMAD4* loss-of-function was associated with a significant increase in proliferation (Figure 2D), confirming our previous observations (Figure 1E). CRISPR-*SMAD4*^R361H^ PDOs became sensitive towards MEK inhibition but not other compartments within the MAPK pathway (Figure 2E and Appendix A). This further strengthens a *SMAD4* loss-of-function driven increased sensitivity towards MEK-inhibitors that may be of use as a predictive biomarker for clinically successful MEK-inhibitor therapy in CRC.

### 3.3. BMP-Pathway Is Activated in Response to MEK Inhibition

To pin down the mechanism underlying the differential MEK-inhibitor sensitivity, we treated the sibling-PDOs and the CRISPR-*SMAD4*^R361H^ PDOs with trametinib (0.03 µM) or vehicle control DMSO (0.3%) for 24 h and performed RNA sequencing. In a Principal Component Analysis, all CRISPR-*SMAD4*^R361H^ PDOs and R1^R361H^ clustered together and were clearly separated from R4^wt^ (Figure 3A). We found 618 genes affected in all *SMAD4*^R361H^ PDOs and 222 unique genes in R4^wt^ (Figure 3B). Gene enrichment analysis of the *SMAD4*^R361H^ models revealed that among the significantly altered biological processes were DNA replication and cell cycle progression consistent with a MEK inactivation-associated growth arrest (Appendix A). Moreover, affected in GO terms were signal transduction, specifically of the TGF-β/BMP signaling, especially in *SMAD4*^wt^ PDOs. Yet, DNA replication or cell cycle progression were not enriched (Appendix A).

As a complementary approach, we subjected trametinib and DMSO-control treated sibling cultures to DigiWest^®^ analysis [32] (Figure 3C,D, Appendix A). All *SMAD4* mutated PDOs clustered together according to their protein profile (Figure 3C). We observed a pronounced decrease in proliferation markers cyclin B1 and Aurora kinase A in *SMAD4*^R361H^ PDOs after 24 h trametinib treatment (Figure 3D). While no differences in post-translational phosphorylation of MAPK pathway proteins, such as MEK and ERK, or p-ERK were observed (Figure 3D) in R4^wt^ PDOs, phosphorylation of mTOR, SMAD5, and SMAD1/5 was higher after treatment with trametinib (Figure 3D). These findings confirm the activation of the TGF-β/BMP pathway, specifically of the BMP branch, in response to exposure to a MEK inhibitor.

### 3.4. SFAB Signature Predicts Sensitivity to MEK Inhibition in CRC PDOs

Next, we evaluated the eligibility of using the *SMAD4* mutational status as a predictive biomarker for MEK-inhibitors in 62 PDOs with known mutational status (Appendix A). In total, 9 PDOs harbor pathogenic *SMAD4* mutations (stated in “The Human Gene Mutation Database”; HGMD^®^ [36,37]). All of these were sensitive to MEK-inhibitor cobimetinib, 6 were sensitive to trametinib, and 5 to selumetinib (Appendix A). From the data generated on the sibling models, we would have expected that only PDOs with loss-of-function *SMAD4* mutations were sensitive to MEK inhibition. However, a significant amount of *SMAD4*^wt^ PDO models were also MEK-inhibitor sensitive (Appendix A).

To solve this conundrum, we looked at additional genes involved in TGF-β/BMP signaling that are frequently mutated in CRC. In addition to *SMAD4*, we identified *FBXW7* (F-box/WD repeat-containing protein 7) [38], *ARID1A* (AT-rich interactive domain-containing protein 1A) [39], and *BMPR2* (bone morphogenetic protein receptor type II) [40] for the marker signature [41,42] (Figure 4A) as our SFAB- (*SMAD4*, *FBXW7*, *ARID1A* or *BMPR2*) signature. We found that at least one loss-of-function mutation is present in one of these genes determines the sensitivity towards MEK inhibition. The frequency of SFAB in the CRC patient cohort (TCGA n = 594; https://www.cbioportal.org, accessed on 18 August 2021) was comparable to the frequency of SFAB in our PDOs (Figure 4B,C). For PDOs with SFAB-signature, we found up to 95% and 70% significant positive prediction for cobimetinib and selumetinib, respectively, and also up to 70% positive prediction for trametinib (Figure 5). Thus, the SFAB signature can be used to reliably predict response to MEK inhibition.

We further investigated if SFAB is superior to the *RAS* status of CRC PDOs in predicting sensitivity to MEK inhibition. About 53% (33/62) of the CRC PDOs carry a pathogenic *KRAS* mutation (Appendix A), which is concordant with published data [43]. Compared to the predictive power of SFAB, the *RAS* mutational status failed to better predict a model’s sensitivity to the MEK-inhibitors cobimetinib, trametinib, and selumetinib (Appendix A). Furthermore, we investigated if the *BRAF* status of CRC PDOs could predict sensitivity to MEK inhibition. About 10% (6/62) of the CRC PDO carry a pathogenic *BRAF* mutation, which is concordant with published data (https://www.cbioportal.org/, accessed on 20 November 2021). *BRAF* status alone failed to better predict sensitivity to the MEK-inhibitors cobimetinib, trametinib, and selumetinib (Appendix A). Apart from one PDO model, all *BRAF* mutant PDOs have an SFAB signature (Appendix A). Therefore, *BRAF* mutations alone are not better than SFAB (Figure 5).

Overall, these findings establish that the SFAB signature predicts sensitivity to MEK inhibition in CRC in a *RAS/BRAF* status-independent manner.

### 3.5. De-Novo Acquired Drug Resistance Can Affect Sensitivity to MEK-Inhibitors

In CRC, 5-FU-based therapies, such as FOLFOX and FOLFIRINOX, are the first-line treatment of choice. MEK-inhibitors are currently not approved for first-line treatment of CRC [44], and thus, they could only be considered for subsequent therapy lines. To recapitulate this clinical situation, we wanted to establish 5-FU resistant sibling- and CRISPR-*SMAD4*^R361H^ PDOs. PDOs were treated with increasing concentrations of 5-FU up to 230 µM (c_max_) [33]. Interestingly, 5-FU resistant lines could only be established from *SMAD4*^R361H^ PDOs. *SMAD4*^wt^ PDO (R4^wt^) did not recover from long-term 5-FU treatment in two independent approaches (data not shown). This is consistent with the previous observations of *SMAD4* functional loss being associated with first-line therapy resistance [13]. 5-FU resistant PDO models and their passage-matched counterparts did not differ in proliferation (Appendix A). As expected, drug screening revealed a significantly increased 5-FU resistance (Appendix A), as well as an increased resistance to all tested targeted agents, including MEK-inhibitors (Appendix A). These observations suggest that *de-novo* acquired drug resistance to 5-FU might share similar mechanisms with resistance to MEK-inhibitors.

## 4. Discussion

The ultimate goal of precision oncology is to identify effective treatments for the individual patient. One route to reaching that goal is linking specific tumor mutations to clinical response to treatment. Combined with chemotherapy, targeted agents are already an important pillar in the standard-of-care treatment of multiple cancers [45]. Among studied examples are the anti-epidermal growth factor receptor (EGFR) antibody cetuximab and the vascular endothelial growth factor (VEGF) antibody bevacizumab [7,46].

The MAPK pathway is dysregulated in more than 30% of human cancers [47]. The aberrant activation of this pathway is usually triggered by gain-of-function mutations in *RAS* and *BRAF* proteins [48]. Mutations in *KRAS* occur in 36–50% of CRC [47] and they interfere with anti-EGFR therapy [49,50]. Only 10–20% of CRC patients are responsive to anti-EGFR treatment, and the *KRAS*/*NRAS* mutational status is the standard-of-care biomarker to exclude patients from therapies based on anti-EGFR anti-bodies [51]. Cetuximab is therefore only approved for *RAS* wild-type tumors. However, 35–45% of these cases still do not respond to this treatment [52,53]. Besides *KRAS*/*NRAS* and *BRAF* mutations, alterations in *PI3KCA* or *PTEN* are likely involved in the resistance to anti-EGFR therapy [50,52,54]. Also, mutations in *FBXW7* and *SMAD4* are prevalent in non-responsive cases [53]. To plan successful therapies for those patients, additional biomarkers are needed. Using a set of biomarkers will help stratify cancer patients into cohorts based on the individual tumor biology [55] rather than only the location of the tumor. We observed that *SMAD4* loss-of-function is associated with differential sensitivity to MEK inhibition (Figure 1).

We were able to introduce the R361H point mutation into R4^wt^ PDOs and generated three CRISPR-PDOs carrying the homozygous *SMAD4*^R361H^ mutation (Figure 2A–C, Appendix A). However, the three CRISPR-PDOs lost the heterozygous *PI3KCA*^H1047R^ mutation after clonal selection (Figure 2C, Appendix A), which was present in both R1^R361H^ and R4^wt^. *PIK3CA* is frequently mutated in CRC with up to 32% [56,57] and is associated with poor prognosis [58,59]. Although the clones lost *PI3KCA*^H1047R^, they showed similar behavior to R1^R361H^ in terms of proliferation time and drug response (Figure 2D–E). Thus, we conclude that this *PI3KCA*^H1047R^ mutation is not involved in the MEK-inhibitor response.

Gene editing confirmed that mutations of *SMAD4* are responsible for the observed differential drug response. We showed that the introduction of the R361H mutation is sufficient to (re-)induce sensitivity towards the MEK-inhibitors cobimetinib, trametinib, and selumetinib (Figure 2).

Further, activation of the TGF-β/BMP signaling pathway, specifically the BMP pathway branch, was observed in *SMAD4*^wt^ PDOs, indicating that BMP signaling is likely a driver of resistance towards MEK inhibition (Figure 3).

We observed an upregulation of mTOR phosphorylation in trametinib-resistant *SMAD4*^wt^ PDO (R4^wt^) and a decrease in phosphorylation for *SMAD4*^R361H^ PDOs (Figure 3). Several mechanisms of MEK-inhibitor resistance are associated with the activation of cellular signaling pathways, such as the PI3K/AKT/mTOR pathway [60]. This finding is consistent with a previously published study showing that activation of mTOR promotes tumor growth and metastases [61].

As previously published, BMP signaling, particularly through SMAD5, can promote cancer cell proliferation and tumor growth [62]. In breast cancer, resistance to paclitaxel can be reversed with the depletion of *SMAD5* [63]. The role of the BMP signaling in CRC is historically controversial, potentially due to the differential status of this signaling in different CRC molecular subtypes [64,65]. There is evidence of BMP pathway activation in the mesenchymal molecular subtype of the CRC, while in other subtypes, it is suppressed [66]. R1^R361H^ and R4^wt^ were derived from a tumor that was described as consensus molecular subtype 2 (CMS2), which is characterized by WNT and MYC activation [15,64]. In addition to CMS classification, the CRC intrinsic subtype B (CRIS-B) is associated with TGF-β-pathway activity, EMT, and poor prognosis [67], which is more applicable to this tumor and correlates with our observations.

From these observations, it is plausible to assume that functional loss of *SMAD4* and thus loss of BMP signaling renders *SMAD4* mutated tumors more sensitive to MEK-inhibitors. However, a loss-of-function mutation or deletion of the *SMAD4* gene alone did not significantly positively predict response to MEK inhibition (Appendix A). By broadening the scope to additional frequently mutated genes involved in TGF-β/BMP signaling [11,38,39,40,41], we identified the novel gene mutational SFAB-signature (*SMAD4*, *FBXW7*, *ARID1A*, or *BMPR2*) (Figure 4). *SMAD4* is considered a signaling mediator of the TGF-β/BMP-pathway, but its role as an important suppressor of CRC progression remains elusive [68]. *SMAD4* loss has been shown to promote chemoresistance in multiple cancer types, including CRC [11,12].

*FBXW7* is a tumor suppressor and regulates the TGF-β/BMP-pathway by targeting for degradation co-repressor TGF-β-induced factor 1 (TGIF1), which recruits specific repressor complexes to SMAD2 [69]. Functional loss of *FBXW7* also promotes chemoresistance in CRC cell lines [70].

*ARID1A* participates in the TGF-β/BMP-pathway as a tumor suppressor that interacts with p53 regulating cyclin-dependent kinase inhibitor 1A (CDKN1A) and SMAD3 transcription and subsequently tumor growth [71]. *ARID1A* has emerged as a candidate “driver gene” tumor suppressor based on its frequent mutations in cancer cells, such as ovarian clear cell and endometroid cancers, as well as CRC [72].

*BMPR2* is directly involved in BMP branch signal transduction in the TGF-β/BMP-pathway [73] and has been reported to play a dual role in regulating tumor growth [74]. In CRC lacking SMAD4, BMPR2 can bind to LIM domain kinase 1 to activate the Rho/Rho-associated protein kinase (ROCK) pathway to promote tumor invasion and metastases [75].

The SFAB signature predicts the sensitivity towards MEK inhibition in CRC PDOs with a high probability (Figure 5). PDO models have been consistently shown to faithfully recapitulate features of the tumor of origin in terms of cell differentiation, genomic background, intra-tumor heterogeneity, and drug response [14,15,76,77,78]. We show that the frequency of SFAB-signature in our PDOs mirrors the expression data of CRC patients (Figure 4). Therefore, it is plausible to assume that the SFAB signature could reliably predict sensitivity to MEK inhibition in CRC patients. Alterations in the TGF-β/BMP pathway due to loss-of-function mutations or deletions affecting *SMAD4*, *FBXW7*, *ARID1A*, or *BMPR2* have previously been shown to correlate with decreased overall survival and to promote chemoresistance in multiple cancer types, such as CRC [11,13], ovarian cancer [79,80,81] and squamous cell carcinoma [82]. This exciting observation suggests that the same alterations that cause resistance to first-line therapies and provide proliferative benefits could at the same time render cancer cells sensitive to certain targeted therapeutics.

In addition to the previously published gene expression signature scores representing MEK-inhibitor sensitivity and MEK-inhibitor adaptive resistance [83,84], SFAB could serve as a predictive biomarker for successful cancer therapy by MEK inhibition.

The present study significantly adds to the concept that multimodal strategies must be considered to deliver effective and specific therapies.

## 5. Conclusions

We identified a novel *RAS/RAF*-independent biomarker signature that could correctly predict the outcome of MEK inhibition *in vitro*. In conclusion, we propose using the SFAB signature as a *RAS*/*BRAF*-independent biomarker for the clinical stratification of patients. Furthermore, we strongly suggest including MEK-inhibitors in the portfolio of therapeutic regimens against CRC.

As the next step to renew the guidelines for CRC treatment, retrospective clinical correlation studies in tumors with known mutation status are ongoing, proofing the clinical benefit of the SFAB signature.

## 6. Patents

These findings were patented in March 2021 (Nr. DE102020102143B3). “A method for determining whether to start or continue a treatment of cancer, a biomarker corresponding to at least one marker gene and the use of a biomarker in the method of the invention.”

## Figures and Tables

**Figure 1 cancers-14-03252-f001:**
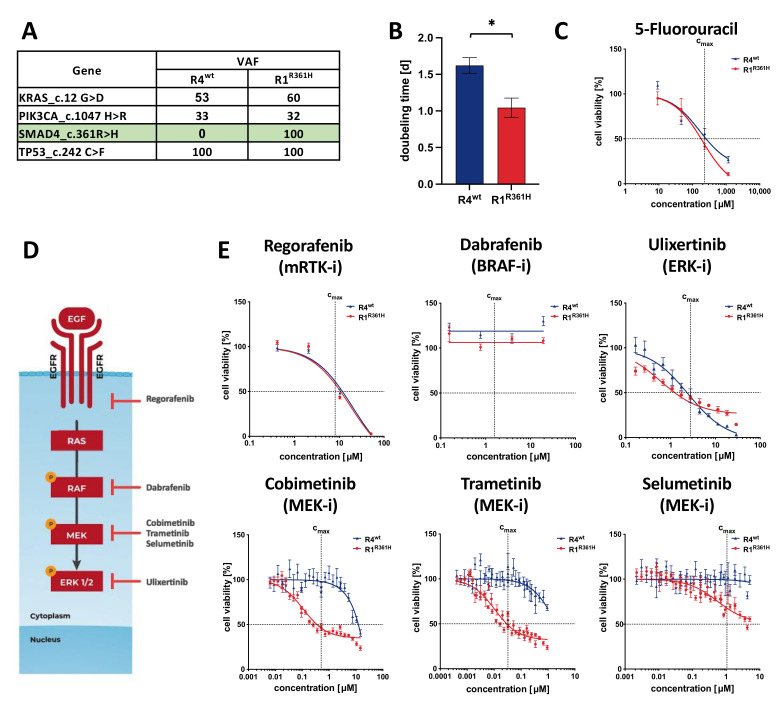
**Single gene alteration leads to differential drug responses.** (**A**) CRC-Panel DNA Sequencing of two representative sibling PDOs differing in SMAD4 mutational status (R4^wt^ and R1^R361H^) (Variant allele frequencies (VAF) are shown). (**B**) Doubling times of R4^wt^ and R1^R361H^. The cell density was measured on day 2, 3, 5, and 7 by measuring cellular ATP content using Cell Titer Glo*^®^* assay. Results from four independent experiments were combined (one-way ANOVA * *p* < 0.05). (**C**) Drug response curves of R4^wt^ and R1^R361H^ treated with 5-Fluorouracil (5-FU). (**D**) Scheme of the MAPK signaling pathway with interruption of signal transduction due to small molecules (mRTK-inhibitor: regorafenib, *BRAF*-inhibitor: dabrafenib, MEK-inhibitors: cobimetinib, trametinib, selumetinib, and ERK-inhibitor: ulixertinib). (**E**) Drug response curves of R4^wt^ and R1^R361H^ treated daily for 4 days with small molecules targeting the MAPK signaling pathway. Cell sensitivity in three biological and four technical replicates was defined according to IC_50_ values in relation to the maximal plasma concentrations (c_max_), which are indicated for each drug with dotted lines. In all panels: Nonlinear curve fit, comparison of best-fit values, extra sum-of-squares F Test *p* < 0.05. (See also Appendix A).

**Figure 2 cancers-14-03252-f002:**
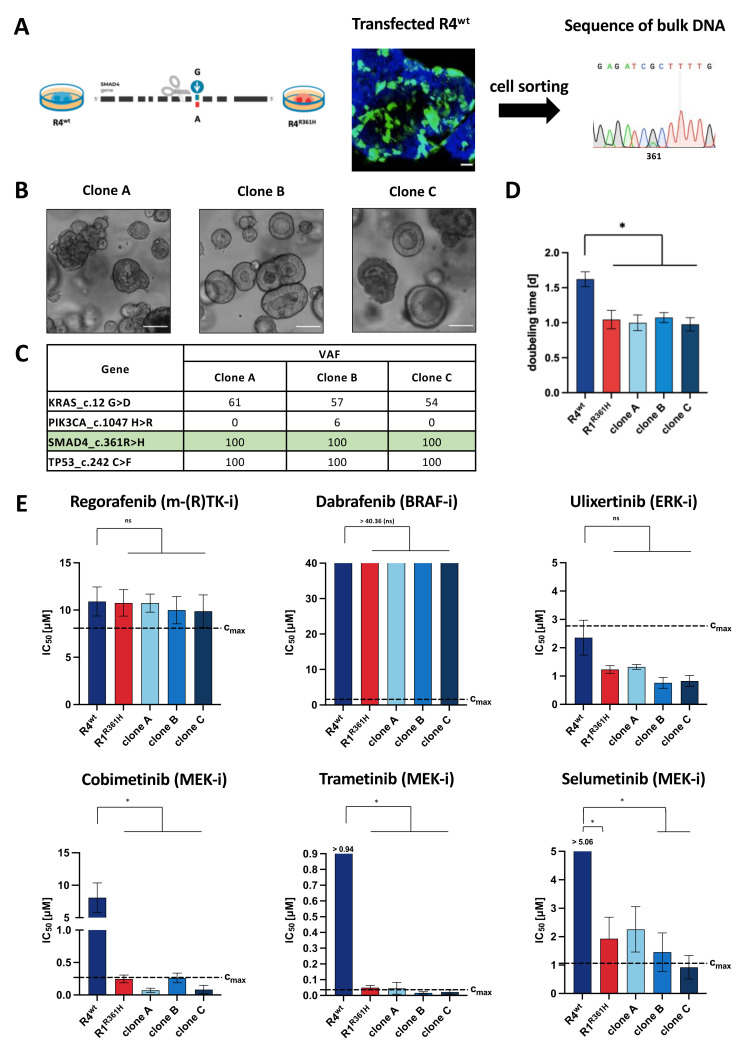
***SMAD4*^R361H^ influences sensitivity to MEK inhibition.** (**A**) Generation of CRISPR-engineered PDOs: transfected R4^wt^ (*SMAD4*^wt^) PDOs were sorted and sequenced as a bulk population. (**B**) Representative images of three single clones (clones A, B, and C) that were expanded for further analysis. (**C**) DNA-panel-sequencing confirmed homozygote *SMAD4*^R361H^ mutation in each CRISPR-*SMAD4*^R361H^ clone (clones A, B, and C). (**D**) Doubling times of R4^wt^, R1^R361H^, and CRISPR-*SMAD4*^R361H^ PDOs (clones A, B, and C). The cell density was measured on day 2, 3, 5, and 7 by measuring cellular ATP content using Cell Titer Glo^®^ assay. Results from four independent experiments were combined (one-way ANOVA * *p* < 0.05). (**E**) IC_50_ values of R4^wt^ and R1^R361H^ and CRISPR-*SMAD4*^R361H^ PDOs (clones A, B, and C each in biological triplicates) treated daily for 4 days with small molecules targeting the MAPK signaling pathway. Achievable maximal plasma concentrations (c_max_) are indicated for each drug with dotted lines. In all panels: ns = not significant; * *p* < 0.05, values as mean ± SEM using one-way ANOVA and Tukey’s test for multiple comparisons. (Scale bars = 100 µm (A and B)). (See also Appendix A).

**Figure 3 cancers-14-03252-f003:**
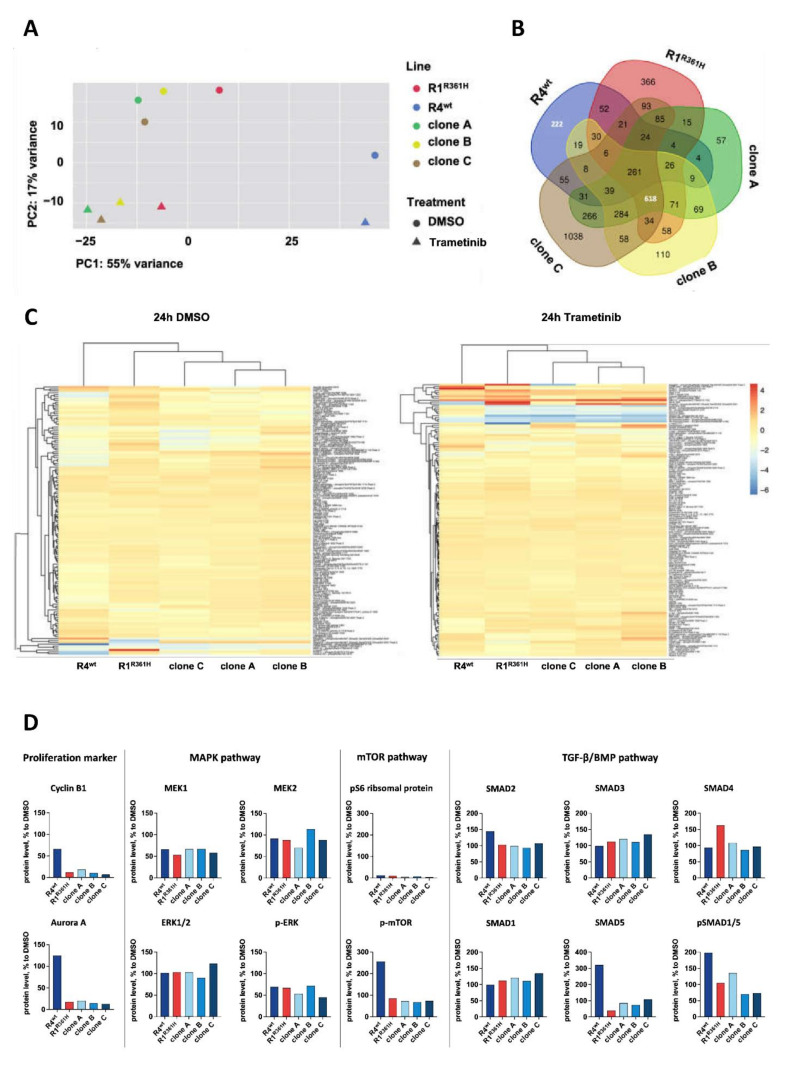
**BMP-pathway is activated in *SMAD4*^wt^ cells in response to MEK-inhibition.** (**A**) Principal Component Analysis of mRNA expression of R4^wt^, R1^R361H^, and CRISPR-*SMAD4*^R361H^ PDOs (clones A, B, and C). The first component on the *x*-axis contains 55% of the variance and classifies the samples into two major groups R1^R361H^/CRISPR-*SMAD4*^R361H^ PDOs (treated with trametinib and untreated) and R4^wt^ (treated with trametinib and untreated). (**B**) Five-set Venn diagram showing unique and overlapping genes from RNA sequencing of R4^wt^, R1^R361H^, and CRISPR-*SMAD4*^R361H^ PDOs (clones A, B, and C). (**C**) Heatmap of differentially expressed total and (phospho-) proteins in R4^wt^, R1^R361H^, and CRISPR-*SMAD4*^R361H^ PDOs (clones A, B, and C) after 24 h-treatment with DMSO (0.3%) and trametinib (0.03 µM) (all normalized over beta-actin, log2 fold change trametinib/DMSO. (**D**) Protein levels [%] involved in proliferation, MAPK pathway, mTOR pathway, and TGF-β/BMP pathway after 24 h-treatment with DMSO (0.3%) and trametinib (0.03 µM). Protein levels were normalized over beta-actin (treatment to DMSO control) in [%] or rather the ratio of phospho-protein level to the total protein level of replicates. (See also Appendix A).

**Figure 4 cancers-14-03252-f004:**
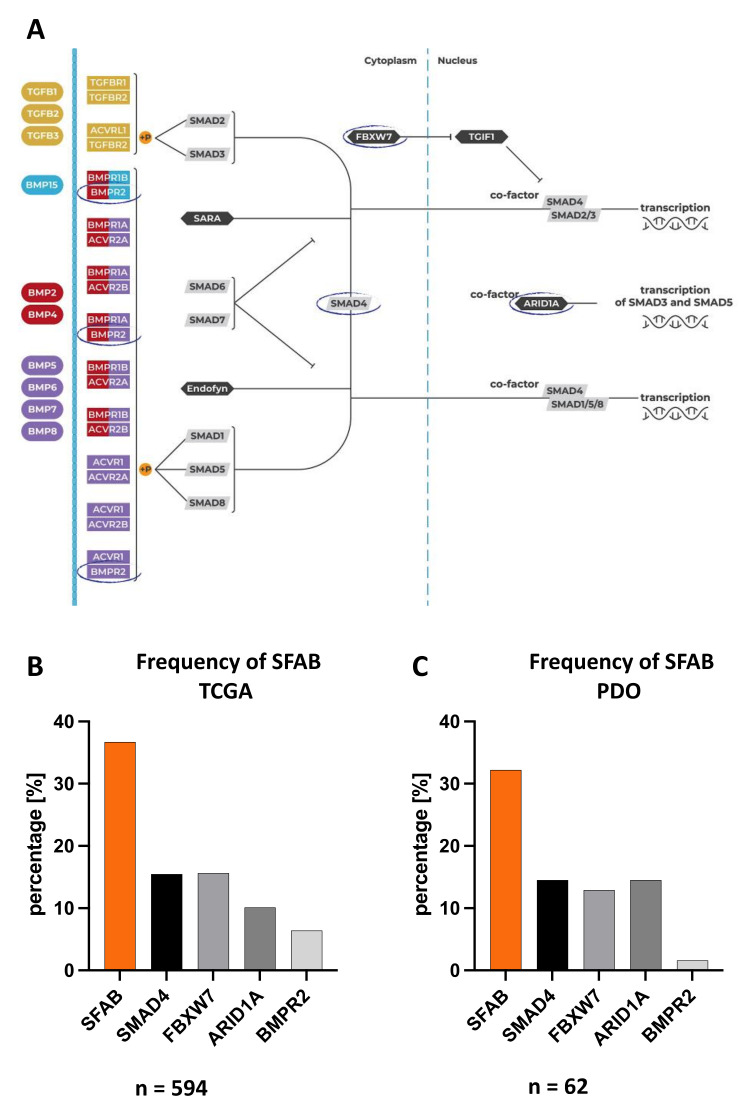
**Four frequently mutated genes in CRC (*SMAD4*, *FBXW7*, *ARID1A,* or *BMPR2*) are involved in the TGF-β/BMP pathway.** (**A**) Scheme of the TGF-β/BMP pathway with four frequently mutated genes in CRC (circled): *SMAD4* = mothers against decapentaplegic homolog 4, *FBXW7* = F-box/WD repeat-containing protein 7, *ARID1A* = AT-rich interactive domain-containing protein 1A, *BMPR2* = Bone morphogenetic protein receptor type II. (**B**) Frequency of SFAB (*SMAD4*, *FBXW7*, *ARID1A*, or *BMPR2*) in CRC patients (TCGA = The Cancer Genome Atlas, PanCancer Atlas colorectal adenocarcinomas (n = 594) (https://www.cbioportal.org, accessed on 18 August 2021)). (**C**) Frequency of SFAB in CRC PDOs (n = 62). (See also Appendix A).

**Figure 5 cancers-14-03252-f005:**
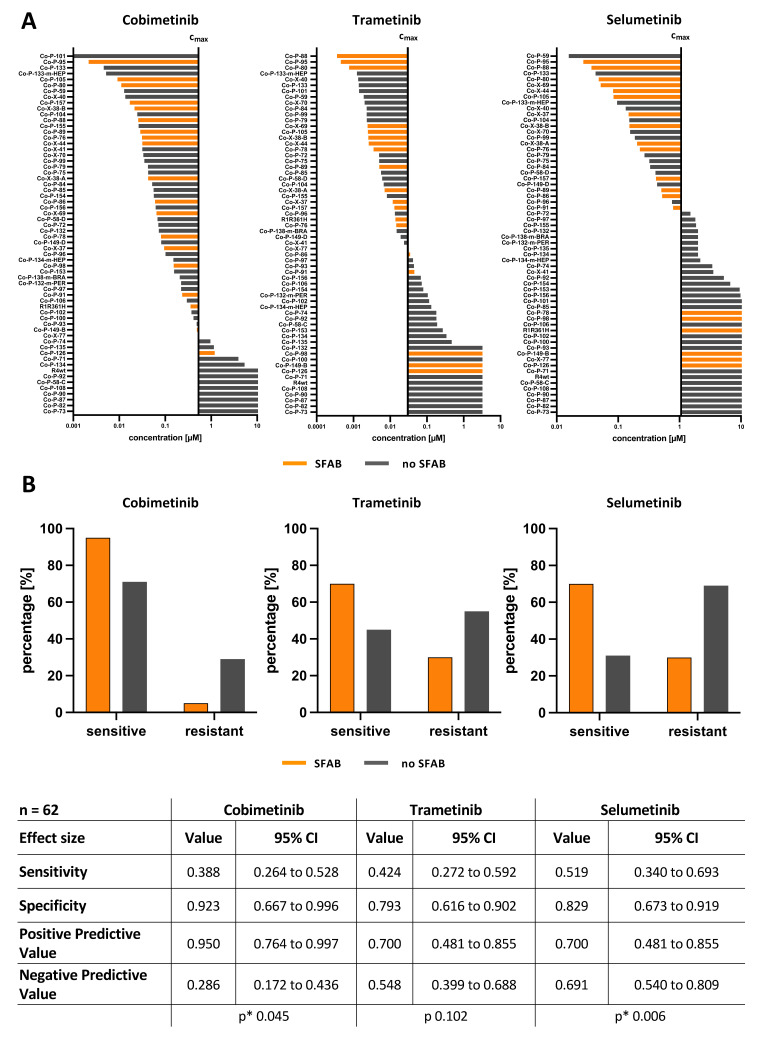
**CRC PDO models with SFAB-** (***SMAD4*, *FBXW7*, *ARID1A*, or *BMPR2*) signature are sensitive to MEK inhibition**. (**A**) Waterfall plots of IC_50_ values of CRC PDO models with known SMAD4 status tested with MEK-inhibitors cobimetinib, trametinib, and selumetinib. PDO models with IC_50_ values below c_max_ are defined as sensitive and PDO models with IC_50_ values above c_max_ are defined as resistant. (Orange bars: PDOs harboring pathogenic mutations in *SMAD4*, *FBXW7*, *ARID1A*, or *BMPR2*; gray bars: PDOs harboring non-pathogenic or no mutations in *SMAD4*, *FBXW7*, *ARID1A*, or *BMPR2*). (**B**) Contingency analysis of the MEK-inhibitors cobimetinib, trametinib, and selumetinib for sensitive and resistant models (n = 62). All calculations were performed on technical replicates of two biological samples each (See also Appendix A). *SMAD4* = mothers against decapentaplegic homolog 4, *FBXW7* = F-box/WD repeat-containing protein 7, *ARID1A* = AT-rich interactive domain-containing protein 1A, *BMPR2* = Bone morphogenetic protein receptor type II.

## Data Availability

The data presented in this study are available in Appendix A.

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
