# Peer review of "A RAS-Independent Biomarker Panel to Reliably Predict Response to MEK Inhibition in Colorectal Cancer"

_cancers, 2022, doi:10.3390/cancers14133252_

Round 1
Reviewer 1 Report
In the present study you address the question of the nature of possible differential MEK inhibitor response of tumor sub clones generated by tumor heterogeneity in colorectal cancer. The aim of the research approach was to identify drugs possibly useful in a refined more personalized medical treatment. Using syngeneic patient-derived organoids, differing only in a CRISPER-engineered SMAD4R361H loss-of-function mutation, the response to diverse drugs in respect to cell survival, RNA and protein expression was analyzed. These syngeneic 3D cell culture models showed striking differences in response for some of the drugs tested on cell viability at clinical related concentrations. The authors correlate pathological SMAD4 loss-of –function alterations to the sensitivity to MEK inhibitor drugs independent of BRAF and RAS gene mutational status. The presented work is a significant building block to understanding the nature of drug response caused by tumor heterogeneity on a single gene level and identifying the pathways possibly involved in drug resistance analyzed at the RNA and protein level. Minor concerns:- Figure 1 C & E (line 261) Font size needs to be enlarged
- Figure “1” (line 330) should be renamed in Figure 3 (as well as cross references in line 309, 310, 319, 320, 322, 325, 455 and 457)
- Figure “1”D (3) (line 329) Font size of the description of the colored bars should be enlarged or replaced by color legend. Enlarge font size of Figure 3 B surrounding the artwork
- Figure “2” (line 380) should be renamed in Figure 4 (as well as cross references in line 478 and 501)
- Figure “3” (line 391) should be renamed in Figure 5 (as well as cross references in line 497)
- Figure “3”A (5) (line 390) Font size (x-axis) should be enlarged
- (line 595) Title of S3 is a bit different from title in the Excel S3 Table, please align the text
- (line 598) Title of S4 is a bit different from title in the Excel S4 Table, please align the text
Author Response
Dear Reviewer, thank you for taking your time reading and commenting on our manuscript. We have happily implemented your suggestions in the revised manuscript. We hope that the conversion of figures now fulfills your needs sufficiently.
In particular:
1) Fig. 1 c and e font size enlarged
2) Fig "1" is renamed in "fig 3"
3) Figure "1"/fig3: b and d: font size enlarged
4) Fig "2" is now "fig 4"
5) Fig "3" is now "fig 5"
6) Fig "3A"/fig 5A font resized
7) Title S3 is aligned accordingly
8) Title S4 is aligned accordingly
On behalf of the authors,
Christian Regenbrecht
Reviewer 2 Report
Pfohl et al. describes a novel colorectal cancer biomarker signature, called SFAB. Patient derived organoids harboring this signature are sensitive to MEK-inhibitors independently from their RAS/RAF status.
The authors investigate and important subject, to personalize CRC treatment. SMAD4 mutations have been associated with decreased overall survival and plague 15% of CRC cases. Importantly, these cancers usually respond poorly to anti-EGFR therapy, independently of their KRAS status. The authors propose that this CRC subtype could respond well to MEK-inhibitors. While the manuscript is overall good, it falls short on establishing solid mechanistic details (e.g., lack of rescue experiments for the CRISPR clones). There is one fundamental observation that would need more spotlight, which is section 3.5 of the results. Here, the authors show that after 5-FU treatment (universal first-line CRC treatment), the resistant SMADR361H mutant PDOs gain resistance against a wide range of inhibitors, including MEK-inhibitors. Therefore, the authors should emphasize more that treatment of SFAB signature CRC with MEK-inhibitors should precede the 5-FU treatment. The authors should test this hypothesis and should have also tried experimentally a combinatory treatment regimen.
Other issues:
11.) Figure hyperlinks were in a lot of cases gibberish and made the reviewing very difficult. For example, line 242: “(错误!未找到引用源。A; 错误!未找到引用源。, and Table S2).” (Fehler! Verweisquelle konnte nicht gefunden werden.)
22.) Figure 3 was referenced and called Figure 1 in the text. The labels on this Figure 3 (page 9) were so small that I was not able to evaluate it. Nothing can be concluded from Figure 3C or Figure 3D this way.
33.) Figure 2 on page 11 should be Figure 4.
44.) Figure 3 on page 12 should be Figure 5.
Author Response
Dear Reviewer,
thank you for taking your time reading and commenting on our manuscript. We have happily implemented your suggestions in the revised manuscript. We hope that the conversion of figures now fulfills your needs sufficiently.
In particular:
1) all figure hyperlinks were removed as suggested
2) Fig "1" is renamed in "fig 3" and font sizes were enlarged
3) Figure "1"/fig3: b and d: font size enlarged
4) Fig "2" is now "fig 4"
5) Fig "3" is now "fig 5"
On behalf of the authors,
Christian Regenbrecht
Reviewer 3 Report
Reviewer comments:
Comments to the Authors
Authors have investigated a SMAD4 loss-of-function mutation, which is associated with chemoresistance and decreased overall survival in CRC. We used patient-derived organoid models (PDOs) of colorectal cancer and CRISPR technology to investigate the impact on drug response. They showed that PDOs with pathogenic SMAD4 mutation are sensitive to MEK-inhibitors. Finally, their finding determined a novel mutational gene signature for predicting positive response towards MEK-inhibitors, regardless of the RAS and BRAF status. The present study helps to personalized cancer therapy by identifying a new biomarker. This study brings novelty to the field to diagnose cancer by identifying a novel mutational signature, predicting positive response towards MEK-inhibitors.
This manuscript is for the most part well written with substantial discussion of results and postulated according to the evidence provided. The manuscript is impressive, and the figures provided were comprehensive but need some improvisation to increase the readability. The references are appropriate and timely.
Minor criticisms
• Please provide the number of replicates in the figure legends and please provide high quality figures. Font size is very small to read especially Figure 1.
• Please undergo a thorough check of the manuscript for typographical and grammatical errors.
Author Response
Dear Reviewer,
thank you for taking your time reading and commenting on our manuscript. We have happily implemented your suggestions in the revised manuscript. We hope that the conversion of figures now fulfills your needs sufficiently.
In particular:
1) Number of replicates is provided in the figure legends
2) Figures revised and font sizes increased
3) language has been edited
On behalf of the authors,
Christian Regenbrecht